# Effects of RhCrOx Cocatalyst Loaded on Different Metal Doped LaFeO₃ Perovskites with Photocatalytic Hydrogen Performance under Visible Light Irradiation

**Tzu Hsuan Chiang \*** [ID], **Gujjula Viswanath and Yu-Si Chen**

Department of Energy Engineering, National United University, No. 2, Lienda, Nan-Shi Li, Miaoli 36006, Taiwan; viswagujjula0@gmail.com (G.V.); chihippig@gmail.com (Y.-S.C.)
**\*** Correspondence: thchiang@nuu.edu.tw; Tel.: +886-3738-2385

**Abstract:** Photocatalytic hydrogen ($H_2$) production by water splitting provides an alternative to fossil fuels using clean and renewable energy, which gives important requirements about the efficiency of photocatalysts, co-catalysts, and sacrificial agents. To achieve higher $H_2$ production efficiencies from water splitting, the study uses different metals such as yttrium (Y), praseodymium (Pr), magnesium (Mg), Indium (In), calcium (Ca), europium (Eu), and terbium (Tb) doped lanthanum iron oxide ($LaFeO_3$) perovskites. They were synthesized using a co-precipitate method in a citric acid solution, which was loaded with the rhodium chromium oxide ($RhCrO_x$) cocatalysts by an impregnation method along with a detailed investigation of photocatalytic hydrogen evolution performance. Photoluminescence (PL) and UV–Vis diffuse reflectance spectra (DRS) measured the rate of electron–hole recombination for $RhCrO_x/Pr-LaFeO_3$ photocatalysts, and X-ray powder diffraction (XRD), scanning electron microscope (SEM), high resolution transmission electron microscope (HRTEM), and X-ray photoelectron spectra (XPS) analyzed their characteristics. The experimental results obtained show that the samples with 0.5 wt.% $RhCrO_x$ loading and 0.1 M Pr-doped $LaFeO_3$ calcined at a temperature of 700 °C ($0.1Pr-LaFeO_3-700$) exhibited the highest photocatalytic $H_2$ evolution rate of 127 μmol $h^{-1}$ $g^{-1}$, which is 34% higher photocatalytic $H_2$ evolution performance than undoped $LaFeO_3$ photocatalysts (94.8 μmol $h^{-1}$ $g^{-1}$). A measure of 20% of triethanolamine (TEOA) enabled a high hole capture capability and promoted $0.1-Pr-LaFeO_3-700$ to get the highest $H_2$ evolution rate.

**Keywords:** metal doped; perovskite; cocatalysts; photocatalytic hydrogen evolution



## 1. Introduction

Hydrogen is an ideal clean fuel of the future and is compatible with current fuel storage and transportation infrastructures, along with suitability for extended periods of storage [1]. Hydrogen gas is supplied to the anode compartment of proton exchange membrane fuel cells (PEMFCs), and is then oxidized to form two protons and two electrons [2]. Compared to conventional fossil fuels used for electricity generation, the PEMFC is considered to be one of the most efficient energy converters and is widely utilized in automotive vehicles [3] due to the low operating temperature, quick start-up capability, rapid response to load changes, and high efficiency [4]. PEMFCs can also be used as a grid-connected electrical generator [2], enabling the clean and efficient production of power and heat from a range of primary energy sources.

The development of efficient processes to utilize naturally available solar energy is an important research direction, and generating hydrogen by splitting water with solar energy has emerged as a strong contender [5]. Photocatalytic hydrogen production by water splitting provides an alternative to fossil fuels using clean and renewable energy. A big challenge faced by the development of a cost-effective and energy-efficient photocatalyst for water splitting is the efficiency of the hydrogen evolution rate. To increase the large-scale production of hydrogen using photocatalytic water splitting, a decrease

in catalyst decay and recombination of electron–hole pairs during operation is required, in addition to reducing the process production cost by developing a cost-effective and energy-efficient photocatalyst [6]. It is also necessary to improve the light absorption, surface reactions, and the hydrogen evolution rate when using a lower percentage of noble metals in the photocatalyst.

Perovskites, used as photocatalysts, have a smaller band gap, and the band edge potentials can be tuned to absorb more of the visible light spectrum [7]. The perovskite materials can be modified by the addition of metal elements to their structure, which can provide the ability to produce hydrogen via photocatalysis in accordance with the needs of specific photocatalytic reactions. $LaFeO_3$ is a small band gap perovskite material that absorbs light in the visible region and demonstrates a reasonably good photocatalytic hydrogen generation [8]. When $LaFeO_3$ is modified with A or B site elements that act as substituents, the general formulas of the resulting chemical compounds are $La_{1-x}A_xFeO_3$ (A = Ca, Sr, Ba, Ce, or other rare earth elements) and $LaFe_{1-x}B_xO_3$ (B = Mn, Co, Cr, or other transition metals), respectively [9]. In addition, $LaFeO_3/g$-$C_3N_4$ heterostructures have been successfully prepared and their photocatalytic hydrogen evolution performance under visible light irradiation was reported [10]. An effective strategy for the fabrication of visible-light-responsive photocatalyst materials for photocatalytic water splitting is to introduce a transition metal dopant into the matrix of the photocatalyst [11]. Suitable transition metal ions doped into photocatalysts can be used to easily tune the electron concentration, mobility, and lifetime of the charge carriers, and effectively alter the electronic structure and band levels of the photocatalyst via the localized or delocalized nature of the doping-induced states [12]. To enhance the photocatalytic $H_2$ evolution, Ru [12] and Rh [13] doped $LaFeO_3$ in glucose aqueous matrices have been studied.

Increasing the activity of a photocatalyst that contains loading metals or metal oxides as cocatalysts dispersed on the surface of the photocatalysts is extremely important. The photogenerated electrons migrate through the photocatalyst to the interface between the cocatalyst and the photocatalyst where they are entrapped by the cocatalyst [14], a process which strongly determines the adsorption and activation abilities of the photocatalytic reactions [15]. In addition, the cocatalysts prevent the adverse electron–hole recombination and accelerate the surface chemical reactions by inhibiting the backward reaction [16]. Noble metals, such as Pt [17], Au [18], Pd [19], and Rh [20] with a higher redox potential and work function and a weaker metal–hydrogen bond strength, are historically the most favorable for photocatalytic hydrogen evolution activity. These noble metal cocatalysts capture the photogenerated electrons to suppress electron–hole recombination and reduce the activation energy for hydrogen production [21]. Similarly, transition metals and their oxides have been employed as cocatalysts to enhance the rate of oxidation [22]. For example, Liu et al. [23] prepared transition metal oxide clusters, including $MnO_x$, $FeO_x$, $CoO_x$, $NiO_x$, and $CuO_x$, that were loaded in situ into $TiO_2$ nanosheets through an impregnation method, which significantly promoted the photocatalytic oxidation of water to $O_2$ compared to $RuO_2/TiO_2$ and $IrO_2/TiO_2$ nanosheets. To achieve higher $H_2$ production efficiencies from water splitting, electron donors are usually required to act as sacrificial reagents to consume holes and prevent the recombination of photoinduced electrons and holes on the photocatalyst surface [24]. The sacrificial agents methanol [25], ethanol [26], triethanolamine [27], disodium ethylenediaminetetraacetic acid [26], and sodium sulfide/sodium sulfite [15] have been used as hole scavengers for photocatalytic $H_2$ evolution from water splitting.

To date, there remains little information on the effects of different metal dopants, cocatalysts, and sacrificial agents on the photocatalytic $H_2$ production of $LaFeO_3$ perovskites. In this study, the novelty of this work lies in the use of Y, Pr, Mg, In, Ca, Eu, and Tb metals doped into $LaFeO_3$, while $RhCrO_x$ was used as the cocatalysts for the photocatalytic $H_2$ evolution from water splitting. The effects of the doped photocatalysts on different sacrificial agents were studied and were found to improve the efficiency of photocatalytic hydrogen evolution under visible light irradiation. The highest hydrogen productivity

obtained 0.5 wt.% $RhCrO_x$ loading and 0.1 M Pr-doped $LaFeO_3$ calcined at a temperature of 700 °C in 20% TEOA solution.

## 2. Results and Discussion

### 2.1. Different Metal Doping in LaFeO₃

To investigate the photocatalytic hydrogen evolution performance of $LaFeO_3$ doped with different metals, 0.1 g of each doped photocatalyst was dispersed in 100 mL of an aqueous solution containing 10 vol% TEOA. The results are shown in Figure 1a, the Pr-$LaFeO_3$ photocatalyst demonstrated a higher photocatalytic hydrogen evolution activity (127 $\mu$mol h$^{-1}$ g$^{-1}$) than the other metal doped and undoped $LaFeO_3$ photocatalysts (94.8 $\mu$mol h$^{-1}$ g$^{-1}$). In addition, the Ca-$LaFeO_3$ photocatalyst displayed no evidence of any hydrogen production.

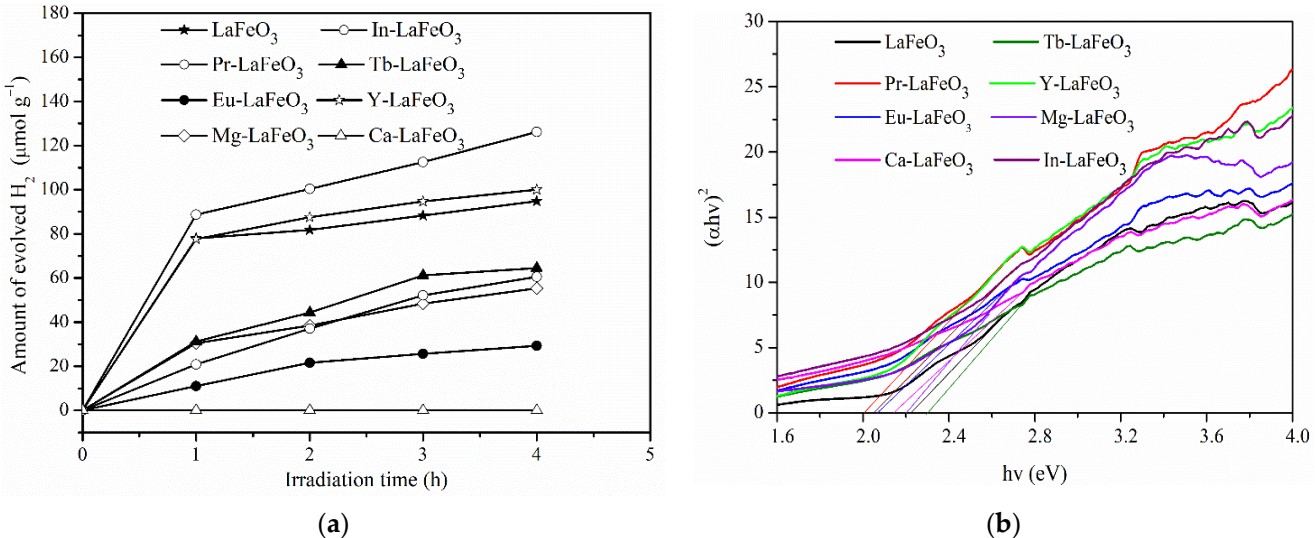

**Figure 1.** (**a**) The hydrogen evolution and (**b**) DRS curves of different metal doped $LaFeO_3$.

The DSR curves of different metal doped $LaFeO_3$ photocatalysts are presented in Figure 1b. The band gap ($E_g$) values were calculated are shown in Table 1. The $E_g$ values for the various metal doped $LaFeO_3$ photocatalysts were smaller than the value for the undoped $LaFeO_3$, with the exception of Tb-$LaFeO_3$, which suggests that metal doping creates a dopant energy level within the band gap of $LaFeO_3$. The lowest band gap was measured for Pr doped $LaFeO_3$ at 2.0 eV, which confirms that the higher activity. Combined with the band gaps calculated from the DSR curves, the valence band energies allowed the determination of the electronic structures and relative band positions of the prepared photocatalysts, as depicted in Table 1 and Figure 2a. Pr-doped $LaFeO_3$ photocatalysts demonstrated a narrower band gap and a more positively shifted valence band as compared to undoped $LaFeO_3$, which allows for more efficient visible-light utilization and charge excitation, while maintaining enough energy for the reduction of water. Consequently, these materials showed an improved visible-light photocatalytic performance for hydrogen production.

**Table 1.** Different photocatalysts and their $E_g$, $E_{CB}$, $E_{VC}$, and crystallite sizes.

| Phtotcatalysts | $E_g$ (eV) | $E_{CB}$ (eV) | $E_{VC}$ (eV) | Crystallite Sizes (nm) |
|---|---|---|---|---|
| LaFeO$_3$ | 2.31 | −0.11 | 2.20 | 28.79 |
| Eu-LaFeO$_3$ | 2.08 | −0.07 | 2.01 | 19.38 |
| Ca-LaFeO$_3$ | 2.17 | −0.10 | 2.07 | 21.39 |
| Pr-LaFeO$_3$ | 2.00 | −0.01 | 1.99 | 16.04 |
| Mg-LaFeO$_3$ | 2.20 | −0.10 | 2.10 | 27.85 |
| Tb-LaFeO$_3$ | 2.30 | −0.17 | 2.13 | 18.31 |
| In-LaFeO$_3$ | 2.05 | −0.04 | 2.01 | 20.10 |
| Y-LaFeO$_3$ | 2.00 | −0.01 | 1.99 | 15.60 |

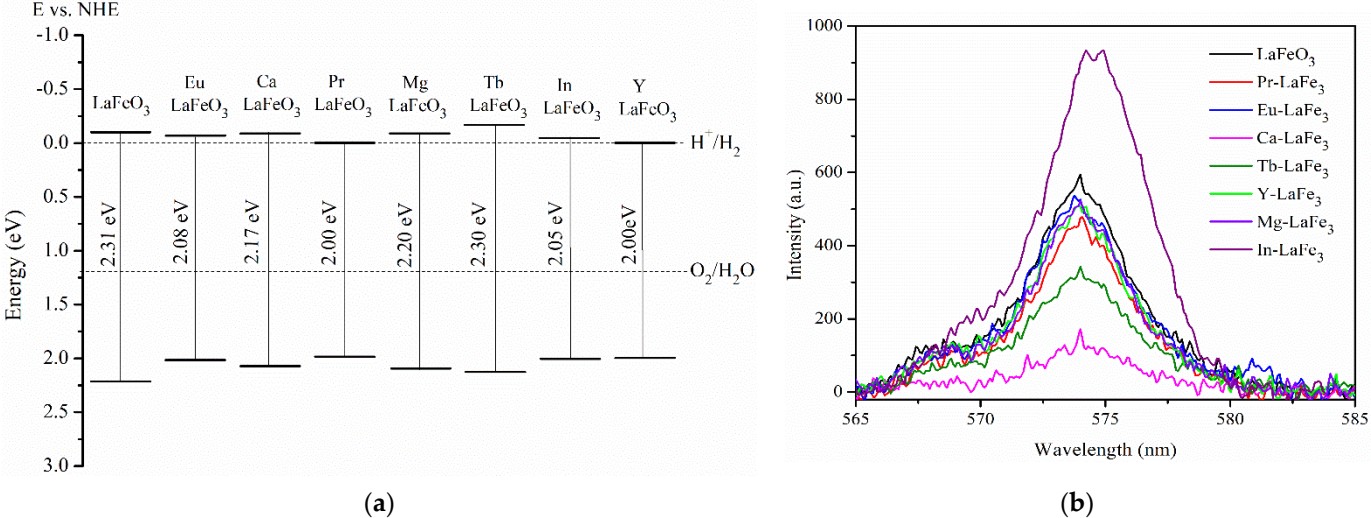

(**a**)　　　　　　　　　　　　　　　　　　　　　　　　　(**b**)

**Figure 2.** (**a**) Band structures and (**b**) photoluminescence spectra of different metal doped LaFeO$_3$ and undoped LaFeO$_3$.

　　　　PL spectroscopy was used to examine the efficiency of trapping, separating, and transferring charge carriers in the semiconductors. In this study, all of the doped LaFeO$_3$ photocatalysts demonstrated similar emission peaks around 574 nm, as shown in Figure 2b. In addition, the different metal doped LaFeO$_3$ were found to have a lower PL intensity than undoped LaFeO$_3$, except for In-LaFeO$_3$, which can be attributed to the presence of slight defects and the introduction of impurity levels within the gaps, both of which encourage charge transfer and reduce the recombination rate of photoinduced electron–hole pairs [28].

　　　　Figure 3 displays the XRD patterns of the different metal doped LaFeO$_3$ samples. The peaks at 2θ values of 22.66°, 32.22°, 39.74°, 46.19°, 57.46°, 67.42°, and 76.61° are in good agreement with the characteristic peaks for the (101), (121), (200), (202) (240), (242), and (204) planes of orthorhombic LaFeO$_3$ (JCPDS no. 371493), respectively. A comparison of the crystallite sizes was accomplished by analyzing the width of the (121) peak as shown Table 1, and it was determined that all of metal had doped into the LaFeO$_3$ structure due to the smaller crystallite size as compared to the undoped LaFeO$_3$. SEM images and EDS analysis after loading with 0.5% RhCrO$_x$ cocatalyst on Pr-LaFeO$_3$ (RhCrO$_x$/Pr-LaFeO$_3$) are shown in Figure 4a,b. The elemental mapping images indicate that the RhCrO$_x$/Pr-LaFeO$_3$ photocatalysts contain La, Fe, Pr, O, Rh, and Cr elements as shown in Figure 4c–h, indicating the successful incorporation of Pr and RhCrO$_x$ cocatalysts into the LaFeO$_3$ photocatalyst. TEM and HRTEM images were utilized to further investigate the structure of Pr-LaFeO$_3$, as shown in Figure 5. The HRTEM image shows lattice fringes with a spacing of 0.278 and 0.227 nm, which are attributed to the (121) and (220) planes of the cubic phase LaFeO$_3$.

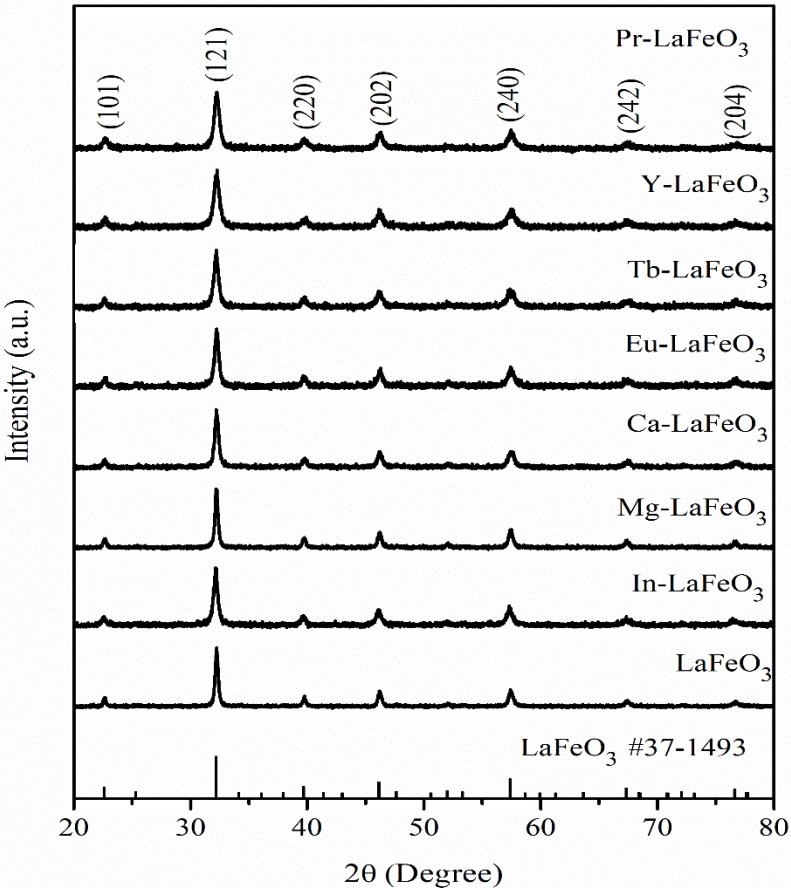

**Figure 3.** XRD patterns of different metal doped LaFeO$_3$ photocatalysts.

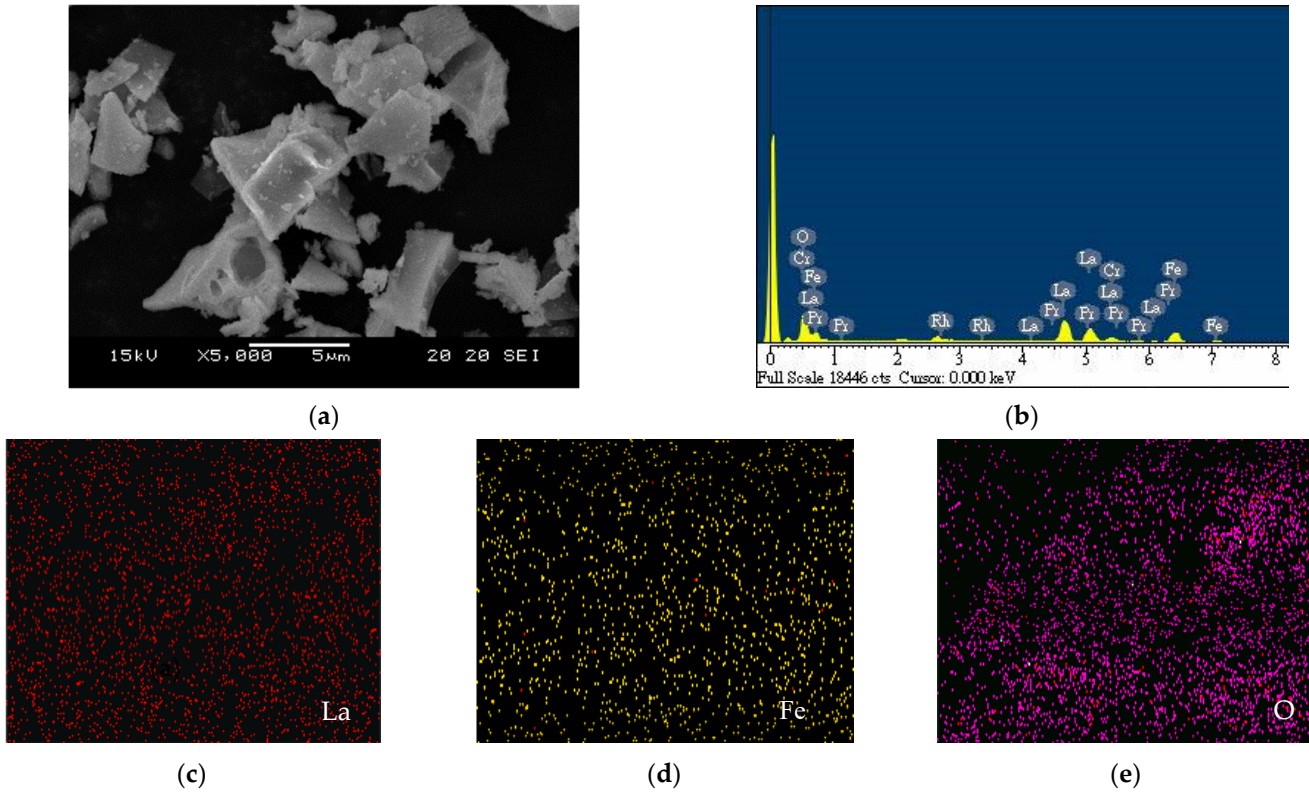

**Figure 4.** *Cont.*

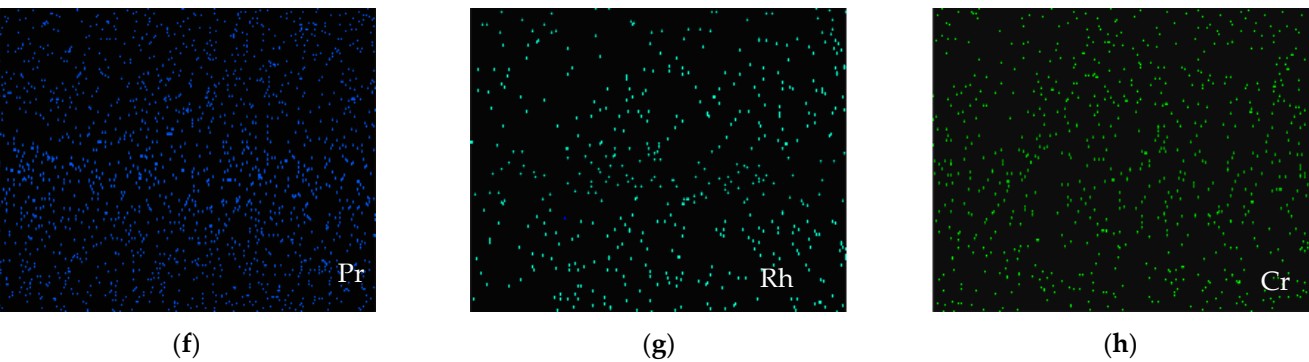

**Figure 4.** (**a**) SEM images, (**b**) EDX, and (**c**–**h**) elemental mapping images of RhCrO$_x$/Pr-LaFeO$_3$ photocatalysts.

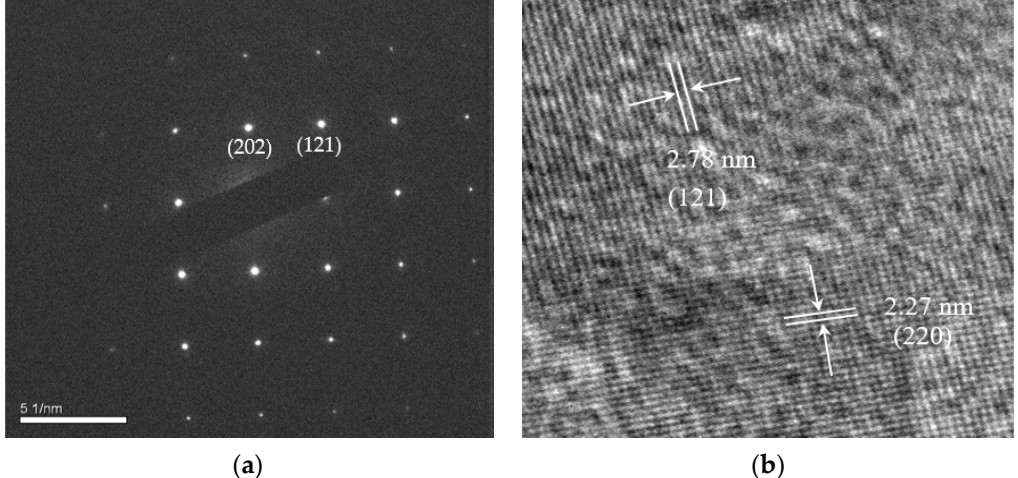

**Figure 5.** (**a**) TEM-associated SAED pattern and (**b**) HRTEM imageof Pr-LaFeO$_3$.

## 2.2. Photocatalytic Activity of Different Amounts of Pr Doping in LaFeO$_3$

Figure 6 shows the photocatalytic H$_2$ evolution rate as a function of the amount of Pr doping in LaFeO$_3$, where Pr-mass doping in each sample was detected using ICP-MS analysis such as 3.5 ppm (0.05 M Pr), 7.9 ppm (0.1 M Pr), 13.1 ppm (0.15 M Pr), 47.8 ppm (0.5 M Pr), and 69.6 ppm (0.7 M Pr). The photocatalytic H$_2$ evolution rate was significantly increased for samples containing Pr at a concentration less than or equal to 0.1 M. When the doping amount of Pr was larger than 0.1 M, the photocatalytic H$_2$ production decreased. In addition, 0.1 M Pr doped in LaFeO$_3$ gave a smaller $E_g$ than at other molar ratios of Pr doping, as shown in Figure 7a. The PL emission spectra of undoped LaFeO$_3$ and various extents of Pr doping in LaFeO$_3$ are shown in Figure 7b. The PL emission intensity at 574 nm for the undoped LaFeO$_3$ is weaker than for Pr doped LaFeO$_3$, and the intensity gradually decreased with increasing amounts of Pr from 0.1 to 0.7 M. These results indicate that the doped more than 0.1 M of Pr may foam more defects such that facilitate the recombination of electrons and holes before redox reaction, thus decreasing photocatalytic activity [29].

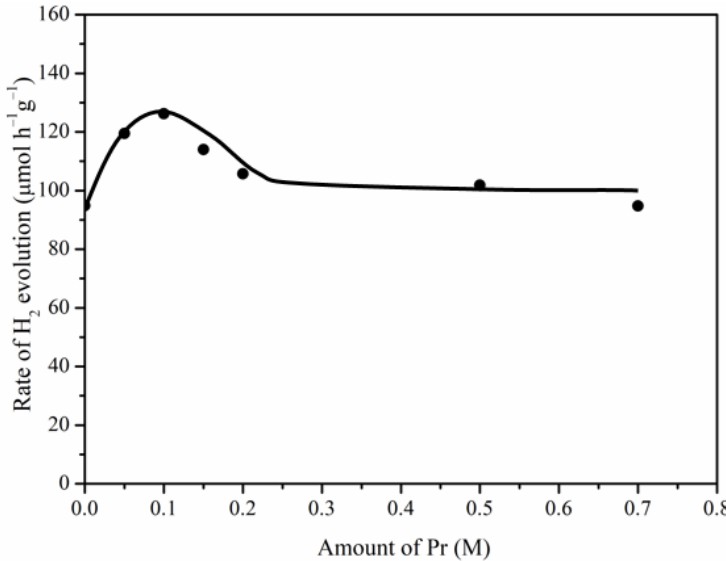

**Figure 6.** Photocatalytic activity of different molar concentrations of Pr doping in LaFeO$_3$ under visible light irradiation.

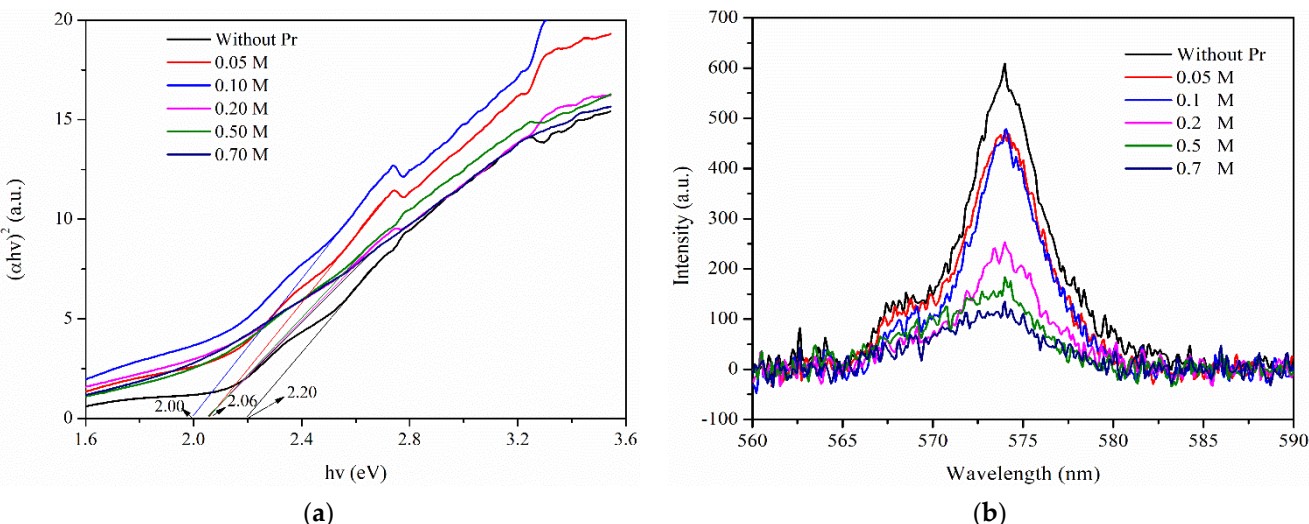

**(a)**                                                                        **(b)**

**Figure 7.** (**a**) UV–Vis DRS spectrum and (**b**) photoluminescence spectra of different concentrations of Pr doping in LaFeO$_3$.

### 2.3. Photocatalytic Activity of Pr-LaFeO$_3$ with Loading of RhCrO$_x$ Cocatalyst

Figure 8 shows the photocatalytic H$_2$ evolution rates obtained from varying amounts of a mixture of Rh and Cr precursor (a fixed 1:1 weight ratio of Rh:Cr) added to Pr-LaFeO$_3$ photocatalysts under visible light irradiation. It was found that the photocatalytic H$_2$ evolution rates increased with higher amounts of loaded Rh and Cr from 0 to 0.5 wt.%. The chemisorption of CO on Rh cocatalysts has been found to inhibit the recombination reaction while maintaining the rate of hydrogen evolution [30]. The results obtained 0.5 wt.% Rh and Cr is the optimal ratio loading on the Pr-LaFeO$_3$ photocatalyst, which suppresses the back reaction lead to produce the highest rate of H$_2$ evolution. However, at greater than 0.5 wt.% Rh and Cr, the H$_2$ evolution rates decreased, presumably because the RhCrO$_x$ blocked active surface sites and shaded on the Pr-LaFeO$_3$ photocatalyst. This was expected, as the loading of an excess of a cocatalyst generally decreases the activity of the photocatalyst. Ran et al. [31] report that the loading of an excess of a cocatalyst generally decreases the activity of photocatalysts by some factors such as covering the surface active sites of the photocatalysts and hindering its contact with sacrificial reagents or water

molecules, shielding the incident light caused inhibition photogenerated electrons and holes inside the photocatalysts, and could act as charge recombination centers. Therefore, 0.5 wt.% RhCrO$_x$ loading on Pr-LaFeO$_3$ was found to give the largest photocatalytic H$_2$ evolution rate, which indicates that 0.5 wt.% of the RhCrO$_x$ cocatalyst is able to capture the photogenerated electrons and suppress the electron–hole recombination.

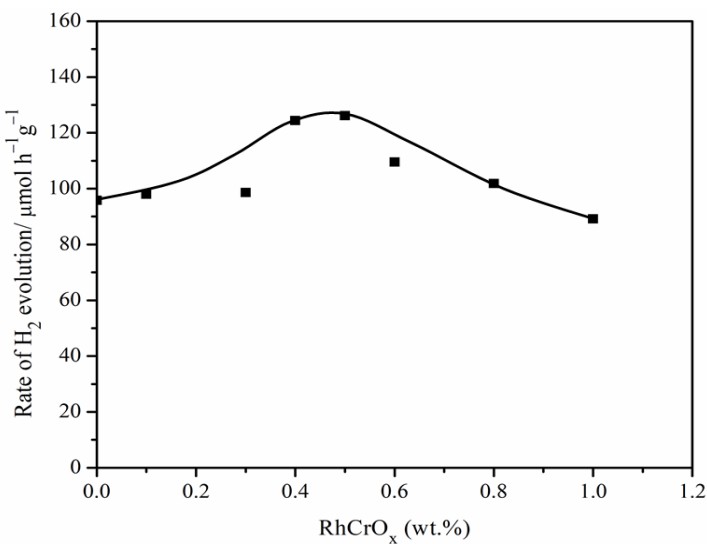

**Figure 8.** Photocatalytic H$_2$ performance of Pr-LaFeO$_3$ photocatalysts modified with different amounts of Rh and Cr cocatalyst.

*2.4. Photocatalytic Activity of Pr-LaFeO$_3$ Prepared at Different Calcination Temperatures*

To investigate the effect of calcination temperature on the photocatalytic activity, Pr-LaFeO$_3$ was calcined at 500, 600, 700, 800, and 900 °C for 4 h with 0.5 wt.% RhCrO$_x$ cocatalyst, and the results are displayed in Figure 9a. It is evident that the photocatalytic H$_2$ evolution rate significantly increases with the calcination temperature from 500 to 700 °C. The highest photocatalytic H$_2$ evolution and the smallest $E_g$ (Figure 9b) were found when Pr-LaFeO$_3$ was calcinated at 700 °C. Unfortunately, the calcination temperature of 500 °C generated a lower photocatalytic H$_2$ evolution rate and a higher PL emission intensity than higher calcination temperatures, as shown in Figure 9c. In Figure 10, the XRD data confirm that the structure of the photocatalyst generated by a calcination temperature of 500 °C did not match with the LaFeO$_3$ structure, leading to the rapid recombination of photo-generated electron–hole pairs. In addition, the intensity of the (121) peak increased with higher calcination temperatures from 600 to 900 °C, which indicates that the crystalline quality increased with higher calcination temperatures, as shown in Figure 10. Generally, the crystalline quality of a photocatalyst affects the charge separation and migration of photogenerated carriers [32]. A high degree of crystallinity usually indicates a smaller quantity of defects and a corresponding increase in catalytic activity. Given that the density of defects impacts the availability of trapping and recombination centers between photogenerated electrons and holes, more defects result in a decrease in the photocatalytic activity [32]. However, in the current study, the photocatalytic H$_2$ evolution rates decreased when the calcination temperature was greater than 800 °C. Therefore, XPS was used to measure the chemical composition of the RhCrO$_x$/Pr-LaFeO$_3$ photocatalysts, including the typical XPS survey spectra of Pr(3d), La(3d), Fe(2p), O(1s), Rh(3d), and Cr(2p), as shown in Figure 11. The binding energies obtained in the XPS analysis were corrected for specimen charging by referencing the C 1s line at 284.5 eV. The binding energies in the XPS spectra for Pr 3d and La 3d did not change when Pr-LaFeO$_3$ was treated at various calcination temperatures. The XPS spectra for Pr 3d can be deconvoluted into three peaks, where the strong peak with a binding energy of 933.4 eV for Pr 3d$_{5/2}$ corresponds to Pr$^{3+}$, and the other two peaks are attributed to Pr$^{4+}$ (937.5 eV) with a shake-off satellite (928 eV),

as shown in Figure 11a. The La 3d doublet was located at 832.5 and 836.1 eV, which was ascribed to $La^{3+}$ and $La^{6+}$ of the $3d_{5/2}$ (Figure 11b). The spin−orbit splitting gap of ~16.8 eV between the $3d_{3/2}$ and $3d_{5/2}$ peaks was indicative of the $La^{3+}$ and $La^{6+}$ state. However, the $Fe^{2+}$ and $Fe^{3+}$ peak areas for Fe 2p varied depending on the calcination temperature, as shown in Figure 11c and Table 2. Two distinct peaks at binding energies of 710.2 and 712.6 eV for Fe $2p_{3/2}$ were observed, which correspond to the characteristic $Fe^{2+}$ and $Fe^{3+}$ ions in the oxide form. The smallest area for $Fe^{2+}$ and the largest area for $Fe^{3+}$ were found at a calcination temperature of 700 °C, which indicates that more $Fe_2O_3$ exists compared to FeO in Pr-LaFeO$_3$. It has been reported that the $Fe_2O_3$ structure has a band gap of 2.17 eV, which can absorb a wider range of the solar spectrum and enhance the photocatalytic $H_2$ evolution [33]. At temperatures greater than 800 °C, the area of $Fe^{2+}$ increased and the area of $Fe^{3+}$ decreased, which indicates that $Fe_2O_3$ was oxidized to FeO and resulted in a decrease in the photocatalytic $H_2$ evolution.

When the Pr-LaFeO$_3$ photocatalysts were calcinated at temperatures greater than or equal to 600 °C, the deconvolution of the O 1s binding energy spectrum results in three peaks that are attributed to lattice oxygen ($O^{2-}$) at 529.4 eV, chemisorbed oxygen ($O^-/O^{2-}$) at 531.6 eV [34], and physically adsorbed oxygen at 533.6 eV [34]. However, the physically adsorbed oxygen peak was absent when the calcination temperature was 500 °C for Pr-LaFeO$_3$, which might indicate a lower crystallization at this temperature.

A similar calcination temperature study was performed after loading RhCrO$_x$ cocatalysts onto the Pr-LaFeO$_3$ photocatalysts at temperatures from 500 to 900 °C. The Rh 3d and Cr 2p XPS binding energies were similar, as shown in Figure 11e,f. The fitting of the data indicates the presence of $Rh^{3+}$ attributable to $Rh_2O_3$ at 308.7 and 313.7 eV, for Rh $3d_{5/2}$ and $3d_{3/2}$, and a satellite peak of $Rh_2O_3$ at 311.1 and 316.1 eV, for Rh $3d_{5/2}$ and $3d_{3/2}$ [35]. In addition, the Cr 2p spectra for all samples were comparable, where the Cr $2p_{3/2}$ signal can be divided into two peaks at 577.4 eV and 580.3 eV, corresponding to $Cr^{3+}$ and $Cr^{6+}$, respectively. The relative amounts of $Cr^{3+}$ and $Cr^{6+}$ ions were analogous, indicating their similar Cr state [36].

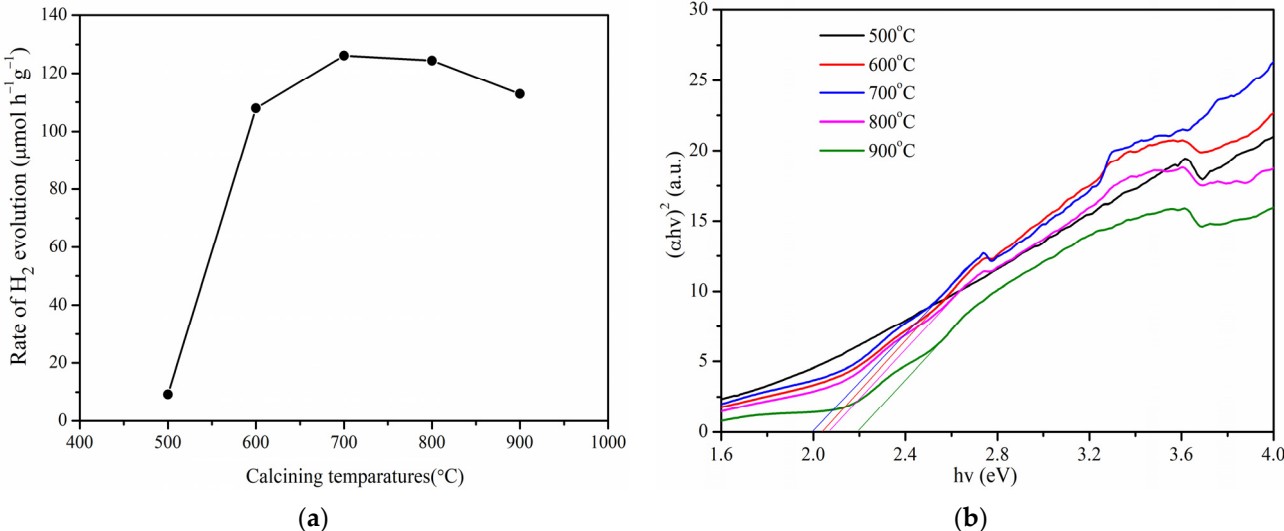

**(a)**          **(b)**

**Figure 9.** *Cont.*

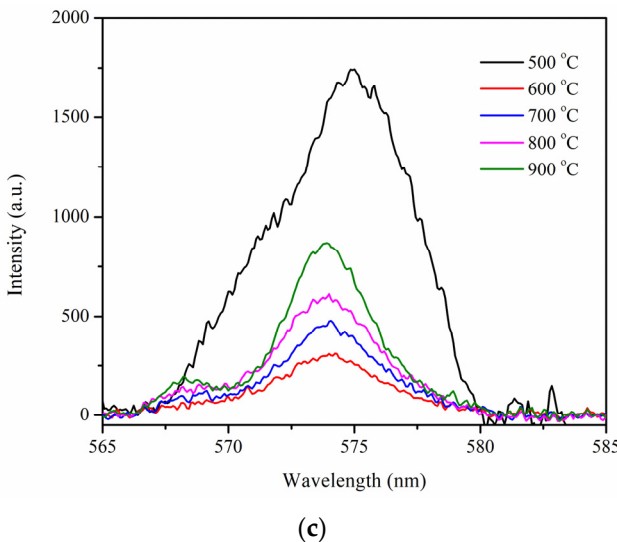

(**c**)

**Figure 9.** (**a**) Photocatalytic $H_2$ evolution; (**b**) UV-DRS spectrum; (**c**) photoluminescence spectra for Pr-LaFeO$_3$ treated at different calcination temperatures with 0.5 wt.% RhCrO$_x$ cocatalyst.

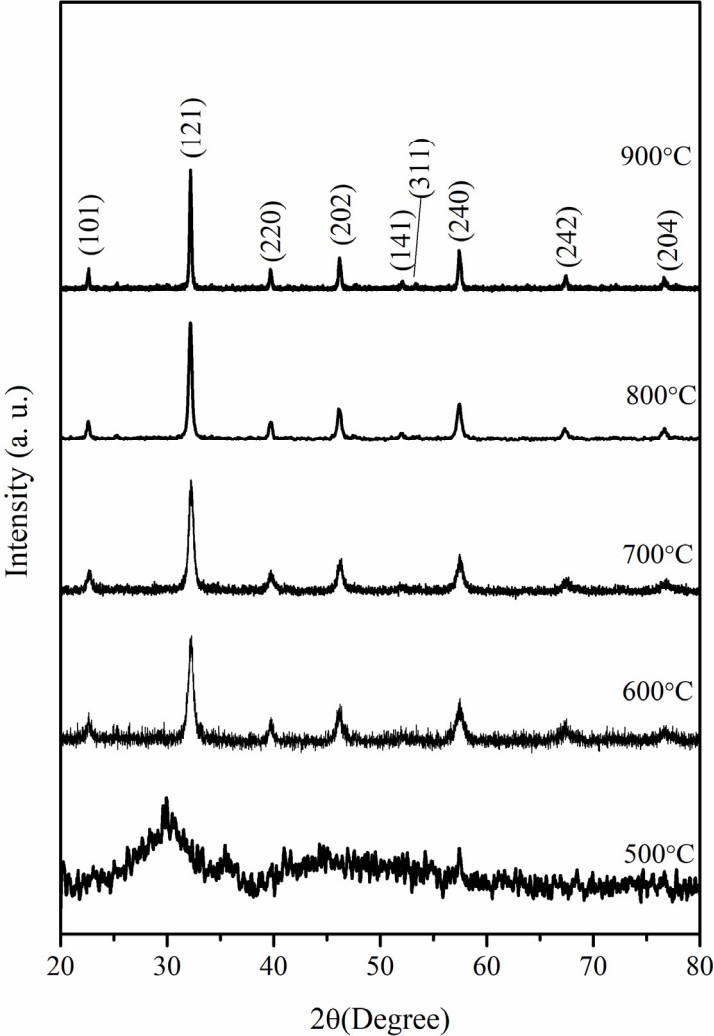

**Figure 10.** XRD patterns of Pr-LaFeO$_3$ after calcination at different temperatures.

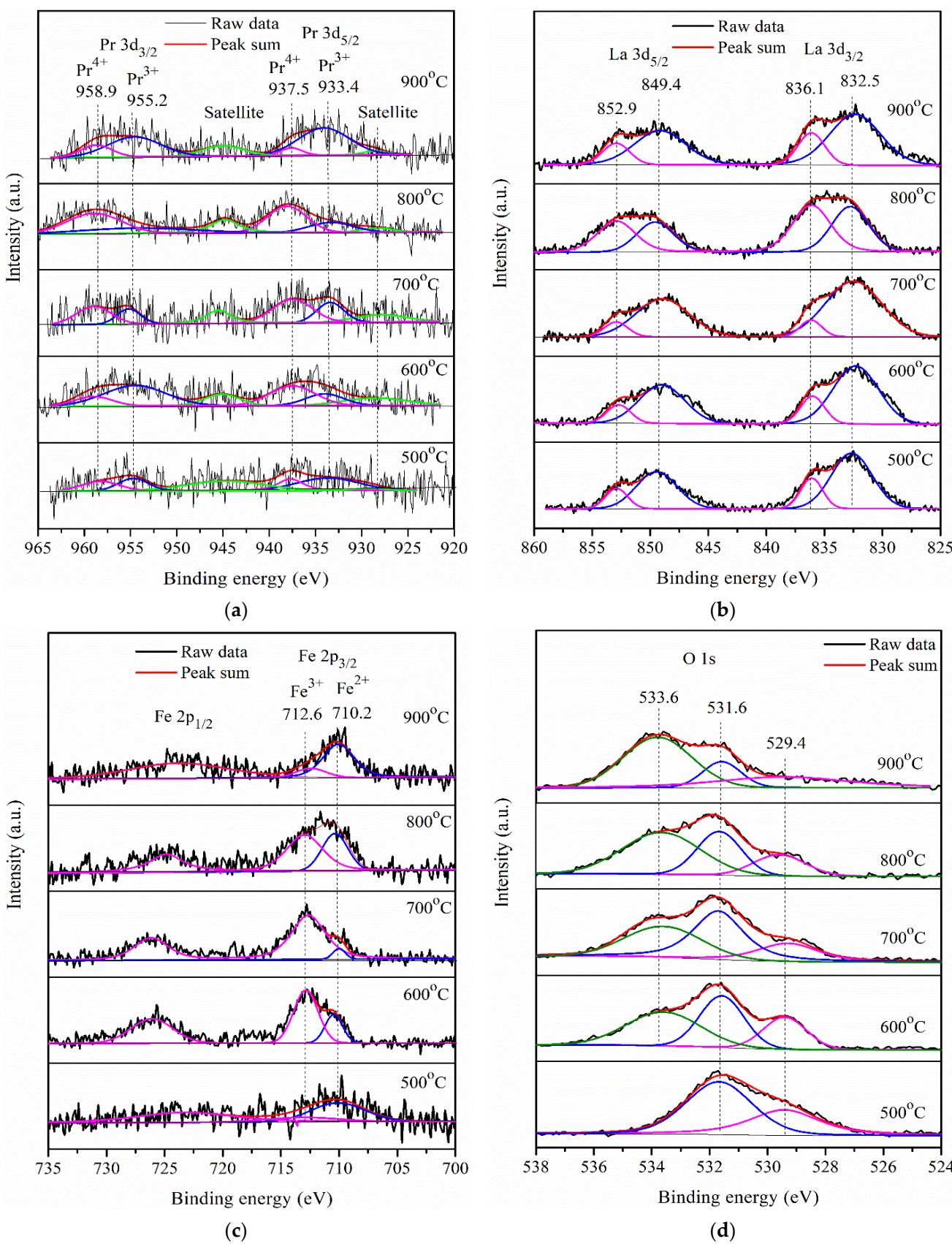

**Figure 11.** *Cont.*

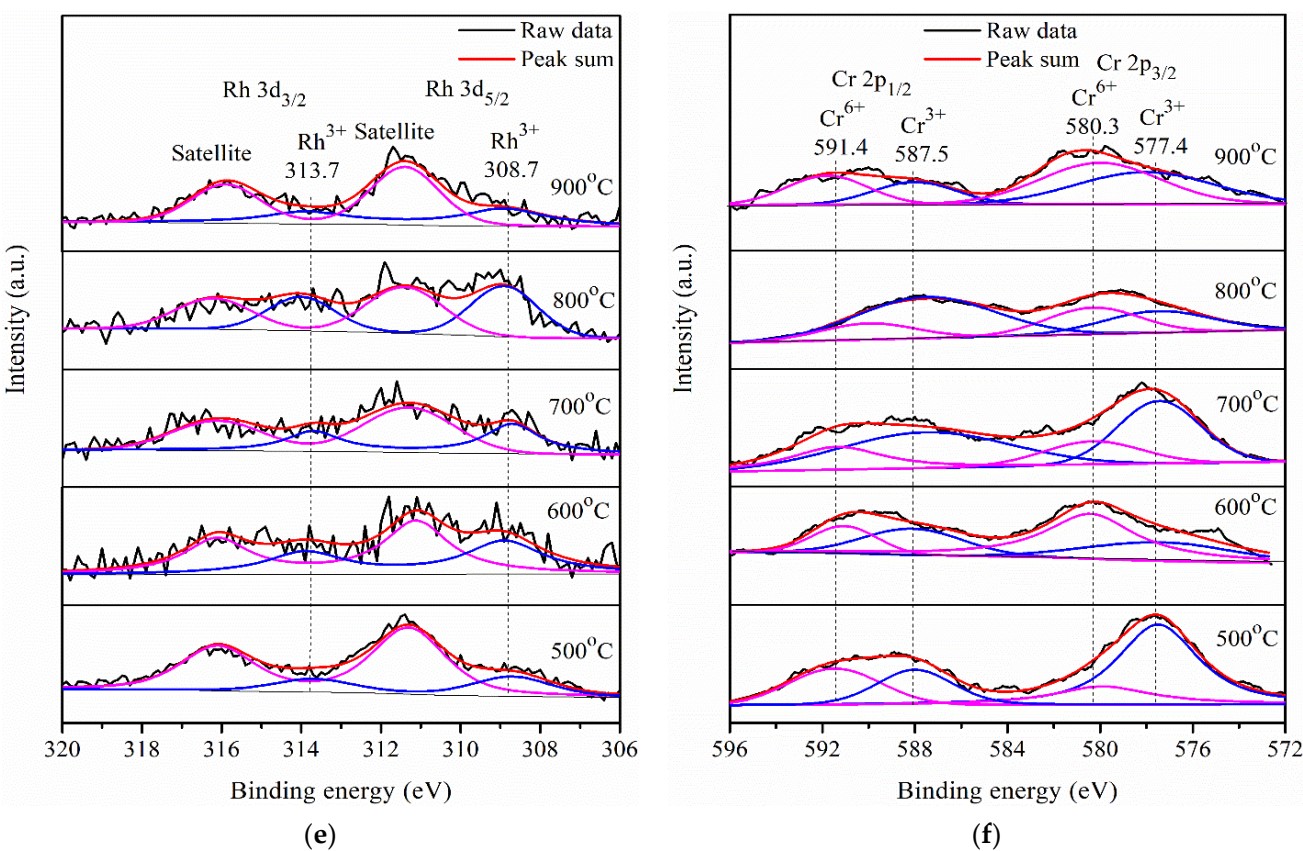

**Figure 11.** XPS spectra of RhCrO$_x$/Pr-LaFeO$_3$ photocatalysts after calcination at different temperatures: (**a**) Pr 3d, (**b**) La 3d, (**c**) Fe 2p, (**d**) O 1s, (**e**) Rh 3d, and (**f**) Cr 2p, and the green, pink and blue fitted curves for different metal.

**Table 2.** Area ratios of Fe$^{2+}$ and Fe$^{3+}$ in the XPS spectra of Pr-LaFeO$_3$ photocatalysts treated at different calcination temperatures.

| Calcining Temperature (°C) | Fe$^{2+}$ (Area %) | Fe$^{3+}$ (Area %) |
|:---:|:---:|:---:|
| 500 | 75.9 | 24.1 |
| 600 | 27.9 | 72.1 |
| 700 | 11.4 | 88.6 |
| 800 | 37.0 | 63.0 |
| 900 | 81.0 | 19.0 |

### 2.5. Photocatalytic Activity of RhCrO$_x$/Pr-LaFeO$_3$ with Different Sacrificial Reagents

Sacrificial agents can be used to increase the photocatalytic activity for water splitting when a photocatalytic reaction is carried out in an aqueous solution containing electron donors or hole scavengers, which prevent electron–hole recombination [24]. The photocatalytic hydrogen evolution of RhCrO$_x$/Pr-LaFeO$_3$ was evaluated with the addition of TEOA, methanol, and ethanol as sacrificial agents with a working volume of 100 mL (90 mL distilled water and 10 mL sacrificial agent), respectively, and the results are shown in Figure 12a. It is clear that RhCrO$_x$/Pr-LaFeO$_3$ has the best photocatalytic H$_2$ production with TEOA compared to the other sacrificial agents. In fact, no H$_2$ evolution was measured for methanol and ethanol, indicating that they are poor hole scavengers for RhCrO$_x$/Pr-LaFeO$_3$. Jones et al. [37] reported that Pd supported on Titania also demonstrated a lower H$_2$ evolution rate in methanol as compared to TEOA. A mechanism for the photo-reforming of methanol is described in Equations (1)–(4) [37]. Methanol has been shown to decarbonylate at ambient temperatures and leave behind CO adsorbed onto the surface of RhCrO$_x$ (Equation (1)), and when CO saturation is achieved, no further reaction (Equation (2)) occurs by band gap excitation in Pr-LaFeO$_3$ to generate electron–hole pairs.

The CO is then removed from the $RhCrO_x$ by the highly electrophilic oxygen species (the hole) in Equation (3). The final step involves the filling of the vacancy ($V_o$) in $Pr\text{-}LaFeO_3$ by water. Thus, the photocatalytic $H_2$ evolution was lower with methanol and ethanol.

$$CH_3OH \rightarrow CO_a + 2H_2 \tag{1}$$

$$O^{2-} + h\nu \rightarrow O^- + e^- \tag{2}$$

$$CO_a + O^- \rightarrow CO_2 + e^- + V_o \tag{3}$$

$$e^- + V_o + H_2O \rightarrow O^{2-} + H_2 \tag{4}$$

Therefore, the choice of TEOA as hole scavenger leads to a higher hole capture efficiency due to the recombination of fewer electrons and holes [38,39] and a lower probability for photooxidation of the semiconductor material [39]. The mechanisms for the photochemical reactions of TEOA are summarized as follows in Equations (5)–(8) [27]:

$$C_6H_{15}NO_3 \rightarrow C_6H_{15}NO_3^+ + e^- \tag{5}$$

$$C_6H_{15}NO_3^+ \rightarrow C_6H_{14}NO_3^\bullet + H^+ \tag{6}$$

$$C_6H_{14}NO_3^\bullet \rightarrow C_6H_{14}NO_3^+ + e^- \tag{7}$$

$$C_6H_{14}NO_3^+ + H_2O \rightarrow C_4H_{11}NO_3 + CH_3CHO + H^+ \tag{8}$$

Figure 12b shows the $H_2$ evolution rate over $RhCrO_x/Pr\text{-}LaFeO_3$ photocatalysts at different concentrations of TEOA. The $H_2$ evolution rate was found to be the highest at 20% of TEOA. The results indicate that at 20% TEOA, the high hole capture capability could effectively accelerate the separation of photo-generated electron–hole pairs upon light irradiation, leading to the highest photocatalytic hydrogen evolution performance. However, when the photocatalysts were tested at 30% TEOA, it is likely that the excessive adsorption of hole scavengers occupied the active sites at the surface of the photocatalyst, leaving no available sites for $H_2$ evolution [40] and generating a corresponding decrease in $H_2$ evolution rates.

In addition, cycling experiments were carried out to determine the stability and reusability of $RhCrO_x/Pr\text{-}LaFeO_3$ for photocatalytic $H_2$ evolution under visible light irradiation. The results shown in Figure 12c reveal that the photocatalytic activity of $RhCrO_x/Pr\text{-}LaFeO_3$ did not suffer any significant loss after four cycles. These results suggest that $RhCrO_x/Pr\text{-}LaFeO_3$ is reasonably stable and could be reused as a photocatalyst with considerable activity.

According to Figure 13, the $Pr\text{-}LaFeO_3$ photocatalysts under visible light irradiation generate electron–hole pairs, where the electrons migrate to the $RhCrO_x$ cocatalysts and participate in the reduction of protons to evolve hydrogen gas. The holes are consumed by the oxidation of the sacrificial agent TEOA, reducing the recombination of the photogenerated charges and improving the photocatalytic $H_2$ evolution.

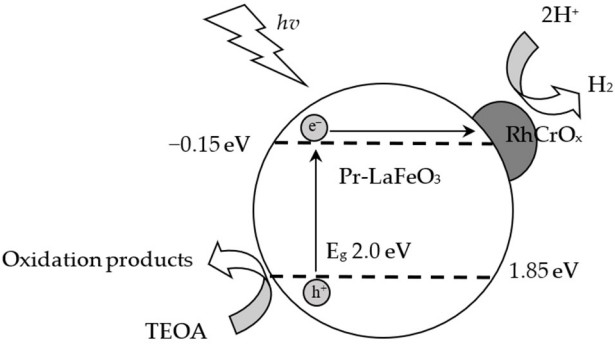

**Figure 12.** (**a**) Photocatalytic activity of $RhCrO_x$/Pr-LaFeO$_3$ with different sacrificial reagents and (**b**) different concentrations of TEOA. (**c**) Cyclic water splitting using $RhCrO_x$/Pr-LaFeO$_3$ photocatalysts.

**Figure 13.** Process of photocatalytic hydrogen evolution over $RhCrO_x$/Pr-LaFeO$_3$ under visible light irradiation with TEOA as the sacrificial reagent.

Table 3 shows that the H$_2$ evolution rates of various LaFeO$_3$-based [8,12,13,41–44] and this study of Pr-LaFeO$_3$. However, they are difficult to compare because the gas evolution rates depend on the different used sacrificial agents, co-catalyst, light source, and the distance between the light source and photocatalysts. Domen's group [45] reported that

the activities of photocatalysts to gas evolution rates in photocatalytic water splitting are extremely difficult to compare the activities measured in different reaction systems because of differences in the photocatalytic reactor systems and in the irradiance of different light sources. The apparent quantum yield (AQY) or apparent quantum efficiency (AQE), can be used as a standard measure of activity. Therefore, the experimental results obtained that the samples with 0.5 wt.% $RhCrO_x$ loading and 0.1 M Pr-doped $LaFeO_3$ calcined at a temperature of 700 °C (0.1-Pr-LaFeO$_3$-700) exhibited the highest photocatalytic $H_2$ evolution rate of 127 $\mu$mol h$^{-1}$ g$^{-1}$, which is 34% higher photocatalytic $H_2$ evolution performance than undoped $LaFeO_3$ photocatalysts (94.8 $\mu$mol h$^{-1}$ g$^{-1}$). We believe that the idealization of this study such as Pr doping and $RhCrO_x$ loading on other $LaFeO_3$-based photocatalysts for previous reports increases their $H_2$ evolution rate at the same of their experimental environment.

**Table 3.** Comparison of $H_2$ evolution rate of $LaFeO_3$-based photocatalysts reported.

| Phtotcatalysts | Sacrificial Agents | Cocatalyst | Light Source | $H_2$ Evolution Rate | Reference |
|---|---|---|---|---|---|
| Pr-doped $LaFeO_3$ | 20% TEOA solution | $RhCrO_x$ | 350 W Xe lamp | 127 $\mu$mol h$^{-1}$ g$^{-1}$ | This work |
| Rh-doped $LaFeO_3$ | Glucose | - | UV-LEDs 10 W, light intensity: 57 mW/cm$^2$) 375–380 nm | 3822 $\mu$mol h$^{-1}$ g$^{-1}$ L$^{-1}$ | [13] |
| Ru-doped $LaFeO_3$ | Glucose | - | UV-LEDs 10 W, light intensity: 57 mW/cm$^2$) 375–380 nm | 875 $\mu$mol h$^{-1}$ g$^{-1}$ | [12] |
| $LaFeO_3$ | Glucose | - | UV-LEDs 10 W, light intensity: 57 mW/cm$^2$) 375–380 nm | 400 $\mu$mol h$^{-1}$ g$^{-1}$ | [41] |
| Ru-$LaFeO_3$/$Fe_2O_3$ | Glucose | | Visible LEDs, 440 nm | 1000 $\mu$mol h$^{-1}$ g$^{-1}$ L$^{-1}$ | [42] |
| Nano $LaFeO_3$ | Ethanol solution | 19 $\mu$L Pt | 400 W tungsten light sources | 3315 $\mu$mol h$^{-1}$ g$^{-1}$ | [8] |
| Polyaniline-coverd $LaFeO_3$ | 10% TEOA solution | 3% Pt | 300 W Xe lamp | 92.4 $\mu$mol h$^{-1}$ | [43] |
| Ternary $LaFe_{0.8}$/$LaCu_{0.2}$ catalysts | 12.5% HCHO solution | - | UV light (cut-off $\lambda$ < 400 nm) 125 W Xe lamp | 250 $\mu$mol h$^{-1}$ g$^{-1}$ | [44] |

## 3. Materials and Methods

### 3.1. Materials

Lanthanum nitrate (La(NO$_3$)$_3$·6H$_2$O), praseodymium acetate hydrate (Pr(CH$_3$CO$_2$)$_3$·xH$_2$O), yttrium (III) nitrate hexahydrate (Y(NO$_3$)$_3$·6H$_2$O), europium (III) acetate hydrate (Eu(CH$_3$CO$_2$)$_3$·xH$_2$O), and sodium hexachlororhodate (III) dodecahydrate (Na$_3$RhCl$_6$·12H$_2$O, Rh 17.1 wt.%) were purchased from Alfa Aesar (Ward Hill, MA, USA). Iron nitrate and citric acid were supplied by Showa Corporation (Gyoda, Japan). Calcium chloride dihydrate (CaCl$_2$·2H$_2$O) and magnesium chloride hexahydrate (MgCl$_2$·6H$_2$O) were supplied by Panreac Química SLU (Barcelona, Spain). Terbium (III) nitrate hexahydrate (Tb(NO$_3$)$_3$·6H$_2$O) and chromium (III) nitrate nonahydrate (Cr(NO$_3$)$_3$·9H$_2$O, 99%) were purchased from Acros organics™ (Carlsbad, CA, USA). Triethanolamine (HOCH$_2$CH$_2$)$_3$N, TEOA) was supplied by Sigma-Aldrich (St. Louis, MO, USA).

### 3.2. Synthesis of Different Metal Doped $LaFeO_3$ Powders

The lanthanum ferrite perovskites were prepared following the method reported by Tijare et al. [8], which involved combining the starting materials, the different doping metals (Pr, In, Mg, Ca, Tb, Eu, and Y), La(NO$_3$)$_3$·6H$_2$O, Fe(NO$_3$)$_3$·9H$_2$O, and citric acid in a molar ratio of 0.1:1:1:4 in 100 mL of water with constant stirring for 60 min, respectively. The mixture was processed in an ultrasonicator at 40 kHz for 120 min. Yellowish-brown precipitates were collected and dried in an oven at 90 °C for 5 h, followed by calcination at 500 °C for 2 h and at 700 °C for 4 h.

### 3.3. RhCrOx/Pr Doped $LaFeO_3$ Preparation

Different molar concentrations (0.1, 0.2, 0.4, 0.5, 0.6, 0.8, and 1.0 M) of Pr doped $LaFeO_3$ photocatalysts were loaded with 0.5 wt.% of the $RhCrO_x$ cocatalyst prepared by combining Na$_3$RhCl$_6$·12H$_2$O and Cr(NO$_3$)$_3$·9H$_2$O in a 1:1 weight ratio by an impregnation method.

### 3.4. Characterization of Photocatalysts

The crystallite size of the different metal doped $LaFeO_3$ samples was determined by X-ray powder diffraction (XRD) using a Rigaku ultimate IV desktop X-ray diffractometer with a Cu K$\alpha$ radiation source at 30 kV and 15 mA and a scanning rate of $5°$ $min^{-1}$. The crystallite sizes ($D_{hkl}$) of the prepared samples were estimated from the line broadening using the Debye–Scherrer equation [46] applied to the (121) peak: $D_{hkl} = K\lambda/B_{hkl} \cos\theta$, where $D_{hkl}$ is the crystallite size in the direction perpendicular to the lattice planes, $\lambda$ is the X-ray wavelength of the Cu Ka radiation ($\lambda$ = 1.5406 Å), $\theta$ is Bragg's angle, $B_{hkl}$ is the pure full width of the diffraction line at half of the maximum intensity, *hkl* are the Miller indices of the analyzed planes, and *K* is a numerical factor frequently referred to as the crystallite-shape factor and is typically 0.9. The morphological properties of the photocatalysts were studied via a scanning electron microscope (SEM) model JOEL JSM-6700F and a high resolution transmission electron microscope (HRTEM) model JOEL JEM-F200 operated at 300 kV. Furthermore, X-ray photoelectron spectra (XPS) were collected to determine the oxidation states and other details of the Pr-$LaFeO_3$ photocatalysts on an ESCA Lab spectrometer with a sigma probe and a mono-chromate Al K$\alpha$ source. The deconvolution of the core-level spectra was performed with the XPS peak fit software. The photoluminescence (PL) spectra were measured at room temperature using a micro-Raman spectrophotometer (UniRaman, ProTrusTech Co., LTD, Tainan, Taiwan) equipped with a laser that operates at an excitation wavelength of 532 nm. The work functions were measured using a photoelectron spectrometer (Model: AC-2) that is an open counter equipped with a UV source manufactured by Riken Keiki Co., LTD, Tokyo, Janpn. Pr elemental analysis was performed using by high-resolution inductively coupled plasma-mass spectrometer (HR-ICP-MS) using Thermo Scientific (Waltham, MA, USA) ELEMENT XR analyzer.

UV–Vis diffuse reflectance spectra (DRS) were measured on a UV–Vis spectrophotometer (U-3900, Hitachi Hight-Tech, Tokyo, Japan) using $BaSO_4$ as a reference. The bandgap energy in the Pr-$LaFeO_3$ photocatalyst was calculated using the formula [47]:

$$\alpha h\upsilon = B(h\upsilon - E_g)^n$$

where $\alpha$ is the optical absorption coefficient, *B* is a constant, $E_g$ is the optical band gap, and *n* is 1/2 or 2 for direct or indirect band gap semiconductors, respectively.

The photocatalysts present on the surface can serve as an electron capture center, providing more active sites. Therefore, it is necessary to determine the conduction band (CB) and valence band (VB) potentials of the photocatalysts. The energy levels were calculated using the following empirical equations [48], and may give a rough estimate of the relative positions of the normal hydrogen electrode (NHE), which could provide a reference to future experimental studies:

$$E_{CB} = X - E_e - 1/2\, E_g$$

$$E_{VB} = E_{CB} + E_g$$

where $E_{VB}$ and $E_{CB}$ are the VB and CB potentials, respectively. Moreover, $E_e$ is the energy of free electrons versus hydrogen (4.5 eV).

*X* is the absolute electronegativity of a pristine semiconductor, and it was calculated using the following equation:

$$X = [\chi\,(A)^a \chi\,(B)^b \chi\,(C)^c]^{(1/a+b+c)}$$

where *a*, *b*, and *c* are the number of atoms in the compounds. Here, given that 0.1 mol of metals are doped in the $LaFeO_3$ and the number of atoms is 0.1, the relative number of atoms for La, Fe, and O is 1, 1, and 3, respectively. For example, for Pr-$LaFeO_3$, $X = [\chi\,(Pr)^{0.1} \chi\,(La)^1 \chi\,(Fe)^1 \chi\,(O)^3]^{(1/0.1+1+1+3)}$. $\chi$ is Mulliken's definition of the electronegativity of a neutral atom [49,50], defined as $\chi = 1/2(A + I)$. *A* is the atom's electron affinity

and *I* is the first ionization energy. A list of the electron affinities and first ionization energies for the various metals used in this study is shown in Table S1.

### 3.5. Calibration Curves for Hydrogen

The photocatalytics of water splitting have been carried out in a photocatalytic reactor. The total hydrogen and oxygen evolution rates were calculated by using these calibration curves. Before making the hydrogen calibration line, the hydrogen peak position and area must be determined using the GC detection signal. The calibration curves for hydrogen and oxygen have been carried out in a photocatalytic reactor (Labsolar-6A, Beijing Perfectlight Technology Co., Ltd., Beijing, China). The reactor was fixed with a condenser, which was further attached to a gas collector and gas chromatograph (GC6890N, Agilent, Santa Clara, CA, USA) with molecular sieve 5A column and thermal conductivity detector (TCD). Argon gas and $N_2$ gas were used as a carrier gas and make-up gas. The $H_2$ calibration equation as $Y = -11.60 + 81.09 \times X$ as shown in Figure S1.

### 3.6. Photocatalytic H₂ Evolution

First, 0.1 g of the photocatalyst was dispersed in 100 mL of an aqueous solution containing 10 vol % TEOA. The reaction temperature was maintained at 20 °C. The hydrogen concentration was monitored using a photocatalytic reactor (Labsolar-6A). A condenser was fixed onto the reactor, which was then attached to a gas collector and a gas chromatograph (GC6890N, Agilent) equipped with a 5 Å molecular sieve column and a thermal conductivity detector (TCD). The entire experimental setup was placed under vacuum using a vacuum pump to remove air. The photocatalytic activity of the photocatalysts was then monitored for hydrogen generation through the photocatalytic water splitting reaction with TEOA as the sacrificial agent, under varying conditions with feasible parametric changes using a visible light source (350 W Xe lamp) with a 400 nm long-pass cut-off filter.

### 4. Conclusions

In summary, $RhCrO_x/Pr$-$LaFeO_3$ photocatalysts were fabricated, characterized, and found to exhibit a significantly improved photocatalytic performance compared to undoped $LaFeO_3$. Pr-doped $LaFeO_3$ photocatalysts demonstrated a narrower band gap and a more positively shifted valence band as compared to undoped $LaFeO_3$, which allows for more efficient visible-light utilization and charge excitation, while maintaining enough energy for the reduction of water. The photocatalysts with 0.1 M Pr doped into $LaFeO_3$ and calcinated at 700 °C ($0.1Pr$-$LaFeO_3$-700) gave the smallest $E_g$ of 2.0 eV, and a low PL emission intensity due to the presence of a high proportion of $Fe_2O_3$ as demonstrated by the Fe 2p XPS data. In addition, 0.5 wt.% $RhCrO_x$ cocatalysts loaded on $0.1Pr$-$LaFeO_3$-700 in 20% TEOA solution showed the highest photocatalytic $H_2$ evolution rate of 127 $\mu$mol h$^{-1}$ g$^{-1}$. These results indicate that this photocatalyst possesses a higher visible light absorption capacity and a lower photogenerated electron–hole recombination rate. Reusability tests indicated that the as-prepared $RhCrO_x/0.1$-$Pr$-$LaFeO_3$-700 photocatalysts are stable and reusable. The choice of 20% of TEOA enabled a high hole capture capability and was found to effectively accelerate the separation of photo-generated electron and hole pairs upon light irradiation, leading to the highest photocatalytic hydrogen evolution performance.

**Supplementary Materials:** The following are available online at https://www.mdpi.com/article/10.3390/catal11050612/s1, Table S1: Electronegativities and the number of atoms for different metals, Figure S1: $H_2$ calibration curve.

**Author Contributions:** Writing—original draft preparation, writing—review and editing, supervision, project administration, conceptualization, and funding acquisition, T.H.C.; investigation, formal analysis and data curation, G.V.; software, Y.-S.C. All authors have read and agreed to the published version of the manuscript.

**Funding:** This research was funded by Ministry of Science and Technology, grant number MST 108-2221-E-239-013.

**Data Availability Statement:** The data presented in this study and supporting information are available.

**Acknowledgments:** We thank Chen, Ching-Shiun at Chang Gung University in Taiwan for helping with PL data measuring.

**Conflicts of Interest:** The authors declare no conflict of interest.

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
