# Peer review of "Effects of RhCrOx Cocatalyst Loaded on Different Metal Doped LaFeO3 Perovskites with Photocatalytic Hydrogen Performance under Visible Light Irradiation"

_catalysts, doi:10.3390/catal11050612_

Round 1
Reviewer 1 Report
The authors report the preparation of Pr-doped LaFeO3 and investigated to influence of several synthetic and operating parameters on H2 photoproduction using RhCrOx as co-catalyst. Some of the results presented may be of interest but the manuscript must be improved before acceptance.
- along the whole manuscript, results should be better discussed in the context of literature.
- Many reports describe the use of pure LaFeO3 or of LaFeO3-based heterostructured photocatalysts for H2 production (the following references should be added: Int. J Hydrogen Energy 2016, 41, 959-966; Int. J. Hydrogen Energy 2020, 45, 17408-17479).
- introduction : the authors must highlight the novelty of their work and the advances made.
- Figure 6 and the related text: the actual content in Pr of LaFeO3 should be determined, for example via ICP analysis.
- the authors must compare the efficiency of Pr-doped LaFeO3 catalysts for H2 photoproduction to other LaFeO3-based catalysts described in the literature and highlight the advances made.
- the quality of the figures can be improved.
Author Response
Response to Reviewer 1 Comments
- along the whole manuscript, results should be better discussed in the context of literature
Reply:
Thank you for your comments. More discussion adds in the context. Please follow red color words.
- Many reports describe the use of pure LaFeO3 or of LaFeO3-based heterostructured photocatalysts for H2 production (the following references should be added: Int. J Hydrogen Energy 2016, 41, 959-966; Int. J. Hydrogen Energy 2020, 45, 17408-17479)
Reply:
Please let me give some comments on these two papers. For Int. J Hydrogen Energy 2016, 41, 959-966, Production of hydrogen from glucose by LaFeO3 based photocatalytic process during water treatment, prepared pure LaFeO3 and produced of H2 in glucose. For Int. J. Hydrogen Energy 2020, 45, 17408-1747, Heterostructured thin LaFeO3/g-C3N4 films for efficient photoelectrochemical hydrogen evolution, it is prepared LaFeO3/g-C3N4 films for efficient photoelectrochemical hydrogen evolution.
Our study is different from these two papers and other publishers. We focused on metal-doped LaFeO3 and RhCrOx as the cocatalyst in the TEOA solution, improve that H2 activity.
- introduction : the authors must highlight the novelty of their work and the advances made
Reply:
We have highlighted the novelty in the introduciton.
- Figure 6 and the related text: the actual content in Pr of LaFeO3 should be determined, for example via ICP analysis.
Reply:
The Pr mass doping LaFeO3 by ICP data shown in the 126 and 127 lines of section 2.2
- the authors must compare the efficiency of Pr-doped LaFeO3 catalysts for H2 photoproduction to other LaFeO3-based catalysts described in the literature and highlight the advances made.
Reply:
We summarized other LaFeO3 based catalysts and compare the efficiency rate of H2 evolution for Pr-doped LaFeO3 in the following Table. However, we are difficult to compare their rate of production H2 because they used different sacrificial agents, co-catalyst, light source, and the distance between light source and photocatalysts. Besides AQY, most papers compared the rate of H2 evolution for without doped and metal-doped.
|
Photocatalysts |
Sacrificial agents |
Co-catalyst |
Light source |
Rate of H2 evolution |
Reference |
|
Pr-doped LaFeO3 |
20%TEOA solution |
RhCrOx |
350 W Xe lamp |
94.8 mmol h−1 g−1 |
This work |
|
Rh-Doped LaFeO3 |
Glucose |
|
UV-LEDs 10W, light intensity:57 mW/cm2) 375–380 nm |
3822 μmol h-1g-1L-1 |
[1] |
|
Ru-doped LaFeO3 |
Glucose |
- |
UV-LEDs 10W, light intensity:57 mW/cm2) 375–380 nm |
875μmol h-1g-1 |
[2] |
|
LaFeO3 |
Glucose |
- |
UV-LEDs 10W, light intensity:57 mW/cm2) 375–380 nm |
400μmol h-1g-1 |
[3] |
|
Ru-LaFeO3/Fe2O3 |
Glucose |
|
visible LEDs, 440 nm |
1000 h-1g-1L-1 |
[4] |
|
Nano LaFeO3 |
Ethanol solution |
19μlPt |
400 W tungsten light sources |
3315μmol h-1g-1 |
[5] |
|
Polyaniline-coverd LaFeO3 |
10%TEOA |
3%Pt |
300 W Xe lamp |
92.4 μmol h-1 |
[6] |
|
Ternary LaFe0.8 / LaCu0.2 catalysts |
12.5% HCHO solution |
|
UV light (cut-off λ < 400 nm) 125W Xe lamp |
250 μmol h-1g-1 |
[7] |
[1] Vaiano,V.; Iervolino,G.; Sannino, D. Enhanced photocatalytic hydrogen production from glucose on Rh-doped LaFeO3, Chem. Eng. Trans., 2017, 60, 235-240.
[2] Iervolino, G.; Vaiano,V.; Sannino, D; Rizzo, L.; Palma, V. Enhanced photocatalytic hydrogen production from glucose aqueousmatrices on Ru-doped LaFeO3, Appl. Catal. B-Enviro. 2017, 207, 182–194.
[3] Iervolino, G.; Vaiano, V.; Sannino, D.; Rizzo, L.; Ciambelli, P. Production of hydrogen from glucose by LaFeO3 based photocatalytic process during water treatment, Int. J. Hydrog. Energy, 41, 2016, 959-966
[4] Iervolino, G.; Vaiano, V.; Sannino, D.; Rizzo, L.; Galluzzi, A.; Polichetti, M.; Pepe, G.; Campiglia, P. Hydrogen production from glucose degradation in water and wastewater treated by Ru-LaFeO3/Fe2O3 magnetic particles photocatalysis and heterogeneous photo-fenton, Int. J. Hydrog. Energy, 43, 2018, 2184-2196.
[5] Tijare, S.N.; Joshi, M.V.; Padole, P.S.; Mangrulkar, P.A.; Rayalu, S.S.; Labhsetwar, N.K. Photocatalytic hydrogen generation through water splitting on nano-crystalline LaFeO3 perovskite, Int. J. Hydrog. Energy, 13, 2012, 10451-10456.
[6] Chen, Z.; Fan, T.; Zhang, Q.; He, J.; Fan, H.; Sun, Y.; Yi, X.; Li, J. Interface engineering: Surface hydrophilic regulation of LaFeO3 towards enhanced visible light photocatalytic hydrogen evolution, J. Colloid Interface Sci. 2019, 536, 105-111.
[7] Li, J.; Pan, X.; Xu, Y.; Jia, L.; Yi, X.; Fang, W. Synergetic effect of copper species as cocatalyst on LaFeO3 for enhanced visible-light photocatalytic hydrogen evolution, Int. J. Hydrog. Energy, 2015, 40, 13918-13925.
- The quality of the figures can be imporved.
Replay:
The quality of all Figures has been improved. We tried our best to make the highest quality of all Figures. However, the word 2016 has maximized the limit 330 ppi quality for Figures.
Reviewer 2 Report
Investigation described in manuscript catalysts-1195459 is concerning were important issue of properties of perovskites doped with different elements for application in hydrogen generation processes. These group of materials posses interesting and promising properties for application in H2 generation processes using solar energy, therefore it may become an important "green" technology.
The manuscript describes interesting and valuable results. However, there are some minor remarks which need to be answered:
1) Is XRD sufficient to determine real chemical composition of obtained perovskites as XRD patterns for each material is quite similar (Figure 3) Chemical analysis of the material could present the real composition of investigated materials.
2) All abbreviations present in the text should be explained, e.g., PL, DSR etc.
3) Figure 6 - trend line should be presented only for positive values of Pr concentrations as it has no physical sense for negative ones.
4) Figure 11 - meaning of violet and blue lines should be explained
5) line 79 - "effects ... was"
Author Response
- Is XRD sufficient to determine real chemical composition of obtained perovskites as XRD patterns for each material is quite similar (Figure 3) Chemical analysis of the material could present the real composition of investigated materials.
Reply:
A small amount of different metal-doped LaFeO3 usually get similar XRD patterns, and the results also confirm by other publishers.For example, Ca doped LaFeO3 (J. Nanosci. Nanotechnol. 2011, 11, 1429–1433), Mg doped LaFeO3 and Zn doped LaFeO3 (ChemSusChem, 2017, 10, 2457-2463) , P doped LaFeO3 (Nano Energy, 2018, 47,199-209, Bi doped LaFeO3 (J. Mater. Sci., 2019, 54, 7460-7468)
However, a comparison of the crystallite sizes for different metal-doped LaFeO3 was accomplished by analyzing the width of the (121) peak as shown Table 1, and it was determined that all of metal had doped into the LaFeO3 structure due to the smaller crystallite size as compared to the undoped LaFeO3.
2. All abbreviations present in the text should be explained, e.g., PL, DSR etc.
Reply:
All abbreviations have modified to full name when they present the first time.
3. Figure 6 - trend line should be presented only for positive values of Pr concentrations as it has no physical sense for negative ones.
Reply:
We change the trend line for positive values.
4. Figure 11 - meaning of violet and blue lines should be explained
Reply:
The meaniing of violet, blue and green lines in XPS are explained in the title of Figure 11.
5. line 79 - "effects ... was”
Reply:
This mistake has been modified.
Round 2
Reviewer 1 Report
From my opinion, the manuscript was not significantly improved compared to the first submission.
In the text, the authors indicate that the H2 evolution rate is of 127 micromol h-1 g-1 using the Pr-doped LaFeO3 photocatalysts. These performances are modest compared to other LaFeO3 based photocatalysts (see the Table in the authors response). In this table, the authors indicate an H2 evolution rate of 94.8 mmol h−1 g−1. This value is not in accordance with that indicated in the main text).
In the present state, I cannot recommend this manuscript for publication in Catalysts. The discussion must be strenghten and the authors must demonstrate the advances made compared to previous reports.
Author Response
Response to Reviewer 1 Comments
According to your suggestion, we have been modified the introduction, methods, results, and conclusions. Please see the red words in the text. And two comments of our reply shown in the following.
- In the text, the authors indicate that the H2 evolution rate is of 127 µmol h-1 g-1 using the Pr-doped LaFeO3 photocatalysts. These performances are modest compared to other LaFeO3 based photocatalysts (see the Table in the authors response). In this table, the authors indicate an H2 evolution rate of 94.8 µmol h−1 g−1. This value is not in accordance with that indicated in the main text)
Reply:
Please accept our apologies to make a mistake for the H2 evolution rate is 94.8 µmol h−1 g−1 in previous comments. We had been modified it, and this Table adds on page 14.
Photocatalysts
Sacrificial agents
Cocatalyst
Light source
H2 evolution rate
Reference
Pr-doped LaFeO3
20%TEOA solution
RhCrOx
350 W Xe lamp
127 µmol h−1 g−1
This work
Rh-Doped LaFeO3
Glucose
UV-LEDs 10W, light intensity:57 mW/cm2) 375–380 nm
3822
μmol h-1g-1L-1
[1]
Ru-doped LaFeO3
Glucose
-
UV-LEDs 10W, light intensity:57 mW/cm2) 375–380 nm
875 μmol h-1g-1
[2]
LaFeO3
Glucose
-
UV-LEDs 10W, light intensity:57 mW/cm2) 375–380 nm
400 μmol h-1g-1
[3]
Ru-LaFeO3/Fe2O3
Glucose
visible
LEDs, 440 nm
1000 μmol h-1g-1L-1
[4]
Nano LaFeO3
Ethanol solution
19μlPt
400 W tungsten light sources
3315 μmol h-1g-1
[5]
Polyaniline-coverd LaFeO3
10%TEOA
3%Pt
300 W Xe lamp
92.4 μmol h-1
[6]
Ternary LaFe0.8 / LaCu0.2 catalysts
12.5% HCHO solution
UV light
(cut-off λ < 400 nm)
125W Xe lamp
250 μmol h-1g-1
[7]
[1] Vaiano,V.; Iervolino,G.; Sannino, D. Enhanced photocatalytic hydrogen production from glucose on Rh-doped LaFeO3, Chem. Eng. Trans., 2017, 60, 235-240.
[2] Iervolino, G.; Vaiano,V.; Sannino, D; Rizzo, L.; Palma, V. Enhanced photocatalytic hydrogen production from glucose aqueousmatrices on Ru-doped LaFeO3, Appl. Catal. B-Enviro. 2017, 207, 182–194.
[3] Iervolino, G.; Vaiano, V.; Sannino, D.; Rizzo, L.; Ciambelli, P. Production of hydrogen from glucose by LaFeO3 based photocatalytic process during water treatment, Int. J. Hydrog. Energy, 41, 2016, 959-966
[4] Iervolino, G.; Vaiano, V.; Sannino, D.; Rizzo, L.; Galluzzi, A.; Polichetti, M.; Pepe, G.; Campiglia, P. Hydrogen production from glucose degradation in water and wastewater treated by Ru-LaFeO3/Fe2O3 magnetic particles photocatalysis and heterogeneous photo-fenton, Int. J. Hydrog. Energy, 43, 2018, 2184-2196.
[5] Tijare, S.N.; Joshi, M.V.; Padole, P.S.; Mangrulkar, P.A.; Rayalu, S.S.; Labhsetwar, N.K. Photocatalytic hydrogen generation through water splitting on nano-crystalline LaFeO3 perovskite, Int. J. Hydrog. Energy, 13, 2012, 10451-10456.
[6] Chen, Z.; Fan, T.; Zhang, Q.; He, J.; Fan, H.; Sun, Y.; Yi, X.; Li, J. Interface engineering: Surface hydrophilic regulation of LaFeO3 towards enhanced visible light photocatalytic hydrogen evolution, J. Colloid Interface Sci. 2019, 536, 105-111.
[7] Li, J.; Pan, X.; Xu, Y.; Jia, L.; Yi, X.; Fang, W. Synergetic effect of copper species as cocatalyst on LaFeO3 for enhanced visible-light photocatalytic hydrogen evolution, Int. J. Hydrog. Energy, 2015, 40, 13918-13925.
2. In the present state, I cannot recommed this manuscript for publicationn Catalysts. The discussion must be strengthen and the authors must demonstrate the advances made compared to previous reports.
Reply:
We apologies again for the difficultly to compare our data to previous reports because the gas evolution rates depend on used different sacrificial agents, co-catalyst, light source, and the distance between light source and photocatalysts. Prof. Domen's group reported that the activities of photocatalysts to gas evolution rates in photocatalytic water splitting are extremely difficult to compare the activities measured in different reaction systems because of differences in the photocatalytic reactor systems and in the irradiance of different light sources. Thus, a measure of photocatalytic performance that is independent of the reactor and light source used is needed. The apparent quantum yield (AQY) or apparent quantum efficiency (AQE), can be used as a standard measure of activity.
Hisatomi, T.; Takanabe, K.; Domen, K. Photocatalytic water-splitting reaction from catalytic and kinetic perspectives, Catal. Lett., 2015, 145, 95-108.
Some previous reports used Pt as a cocatalyst, and we also used Pt cocatalyst and loading on Pr-LaFeO3, which obtained lower H2 evolution as shown in the following Figure. Therefore, the experimental results obtained that the samples with 0.5 wt.% RhCrOx loading and 0.1 M of Pr doped LaFeO3 calcined at a temperature of 700°C (0.1Pr-LaFeO3-700) exhibited the highest photocatalytic H2 evolution rate of 127 µmol h−1 g−1, which is a higher 34% of photocatalytic H2 evolution performance than undoped LaFeO3 photocatalysts (94.8 µmol h−1 g−1).
We believe that the idealization of this study such as Pr doping and RhCrOx loading on other LaFeO3-based catalysts for previous reports increases their H2 evolution rate at the same of their experimental environment.

Round 3
Reviewer 1 Report
The manuscript was improved by the authors and can be accepted by Catalysts.
The following corrections can be done at the proof stage :
- Figure 5 : (a) TEM-associated SAED pattern and (b) HRTEM image of ...
- Table 3 : photocatalyst